# ENDIVE: A CROSS-DIALECT BENCHMARK FOR FAIRNESS AND PERFORMANCE IN LARGE LANGUAGE MODELS

## ABSTRACT

The diversity of human language, shaped by social, cultural, and regional influences, presents significant challenges for natural language processing (NLP) systems. Existing benchmarks often overlook intra-language variations, leaving speakers of non-standard dialects underserved. To address this gap, we introduce **ENDIVE** (**En**glish **Div**ersity), a benchmark that evaluates five widely-used large language models (LLMs) across tasks in language understanding, algorithmic reasoning, mathematics, and logic. Our framework translates Standard American English datasets into five underrepresented dialects using few-shot prompting with verified examples from native speakers, and compare these translations against rule-based methods via fluency assessments, preference tests, and semantic similarity metrics. Human evaluations confirm high translation quality, with average scores of at least 6.02/7 for faithfulness, fluency, and formality. By filtering out near-identical translations, we create a challenging dataset that reveals significant performance disparities—**models consistently underperform on dialectal inputs compared to Standard American English. ENDIVE** thus advances dialect-aware NLP by uncovering model biases and promoting more equitable language technologies.

## 1 INTRODUCTION

Language diversity, shaped by social and cultural factors, presents significant challenges for NLP systems. While English serves as a global lingua franca, its dialects exhibit substantial variation that often goes unaddressed in language technologies Chambers & Trudgill (1998). This oversight perpetuates discrimination against dialect speakers in critical domains like education and employment Purnell et al. (1999); Hofmann et al. (2024a), exacerbated by LLMs' predominant focus on Standard American English (SAE) Blodgett et al. (2016).

Recent studies reveal systemic biases in LLM processing of non-standard dialects Fleisig et al. (2024); Resende et al. (2024)—from toxic speech misclassification of African American Vernacular English tweets Sap et al. (2019) to parsing errors in Chicano and Jamaican English Fought (2003); Patrick (1999). Similar issues plague Indian and Singaporean English due to morphological divergences Kachru (1983); Gupta (1994), highlighting an urgent need for inclusive NLP systems Ziems et al. (2022).

Existing benchmarks like GLUE Wang et al. (2019) and SuperGLUE Wang et al. (2020) fail to capture dialect variation, while specialized datasets (SVAMP, MBPP, FOLIO) Patel et al. (2021); Austin et al. (2021); Han et al. (2024) remain SAE-centric. While frameworks like Multi-VALUE Ziems et al. (2023; 2022) address dialect representation through rule-based lexical substitutions, their synthetic approach fails to capture authentic syntactic patterns. This limitation is particularly acute in reasoning tasks, where surface-level translations preserve logical meaning but lose dialect-specific pragmatic markers essential for fair evaluation.

To address these gaps, we introduce **ENDIVE** (**En**glish **Diversity**), a benchmark that evaluates five LLMs across 12 natural language understanding (NLU) tasks translated into five underrepresented dialects selected for their linguistic distinctiveness and sociocultural significance:

- **African American Vernacular English (AAVE)**: 33M speakers with distinct syntax/phonology Lippi-Green (1997)

- **Indian English (IndE)**: 250M speakers blending local/colonial influences Kachru (1983)

- **Jamaican English (JamE)**: Diaspora language with mesolectal variation Patrick (1999)

- **Chicano English (ChcE)**: Spanish-influenced variety in US Hispanic communities Fought (2003)

- **Colloquial Singaporean English (CollSgE)**: Multicultural creole with Asian substrates Platt & Weber (1980)

Our methodology combines linguistic authenticity with strategic filtering to create robust dialect evaluations. Using verified text samples in the target dialects from eWAVE Kortmann et al. (2020) for few-shot prompting, we translate SAE datasets into target dialects while preserving sociolinguistic nuance. To eliminate superficial transformations, we apply BLEU-based filtering Papineni et al. (2002), removing translations with scores $\geq 0.7$ against their SAE sources—retaining only substantive linguistic variations that challenge LLMs' dialect understanding. We compare our translations against Multi-VALUE's rule-based translations Ziems et al. (2023) through fluency assessments, semantic similarity metrics, and LLM preference tests. Additionally, we have native speakers assess our translations to ensure linguistic authenticity and original content meaning are preserved across all five dialects.

**Our Contributions:**

1. **Public Benchmark**: Curated challenging dialectal variants across 12 reasoning and natural language understanding tasks, validated via multiple metrics and human evaluation.

2. **Cross-LLM Evaluation**: Comprehensive testing of 5 LLMs (GPT-4o, GPT-4o mini, Claude-3.5-Sonnet, Deepseek-v3, LLaMa-3-8b) revealing consistent performance disparities between SAE and dialectal inputs, using chain-of-thought (CoT) and zero-shot prompting.

## 2    RELATED WORK

**Dialectal Diversity.** Addressing dialectal diversity in NLP remains a significant challenge due to inherent linguistic variations shaped by social and cultural contexts. Early research identified systemic biases in language models against non-standard dialects such as AAVE, highlighting issues like the misclassification of AAVE tweets as toxic and difficulties in syntactic parsing Sap et al. (2019); Jørgensen et al. (2015). Recent studies extend these findings to modern LLMs, revealing persistent dialect prejudice in evaluations related to employability, criminality, and medical diagnoses Hofmann et al. (2024b); Fleisig et al. (2024); Blodgett & O'Connor (2017).

**Benchmarking Approaches and Hybrid Methodologies.** Dialect robustness is primarily evaluated using two approaches. The first relies on rule-based lexical substitutions—exemplified by VALUE and Multi-VALUE Ziems et al. (2022; 2023)—which are scalable but often miss nuanced, context-dependent features (e.g., AAVE's habitual "be" Green (2002); Lippi-Green (1997) or Chicano English's Spanish-influenced prosody Fought (2003); Santa Ana (1993)). The second employs human-annotated translations (e.g., ReDial, AraDiCE Lin et al. (2025); Mousi et al. (2024)), ensuring authenticity but typically focusing on a single dialect. Recent hybrid methodologies combine automated translation with native speaker validation to balance scalability and authenticity. For example, AraDiCE integrates automated translations with post-edits for Arabic dialects, while AAVENUE Gupta et al. (2024) provides human-validated evaluations for AAVE in NLU tasks. These hybrid approaches offer a more robust framework for comprehensive dialect fairness evaluations.

**Sociolinguistic Impact and Real-World Discrimination.** Beyond technical benchmarks, sociolinguistic studies have linked LLM biases to real-world discrimination—such as housing denials for AAVE speakers Hofmann et al. (2024b); Purnell et al. (1999) and biased criminal justice assessments Fleisig et al. (2024). Multilingual initiatives like LLM for Everyone Cahyawijaya (2024) advocate for continuously fine-tuning models to better serve underrepresented languages. Our approach reflects this tuning perspective by using human-guided few-shot prompting with authentic

linguistic examples Kortmann et al. (2020); Platt & Weber (1980) to generate dialect-specific translations that effectively "tune" the input data, ensuring that the unique features of underrepresented dialects are accurately captured. This alignment helps mitigate model biases and promotes more equitable language technologies.

**Remaining Gaps and Our Contribution.** Although prior work has deepened our understanding of dialect biases in NLP, significant gaps remain in developing comprehensive, multi-dialect benchmarks that integrate authentic linguistic features. ENDIVE addresses these gaps by providing a robust benchmark that combines both automated and human-validated translation methods, thereby fostering more equitable language technology development.

## 3 DATASET

### 3.1 DATASET OVERVIEW

ENDIVE is a benchmark designed to evaluate the reasoning capabilities of LLMs across five underrepresented dialects. The benchmark is curated from 12 established datasets, spanning four core reasoning categories: Language Understanding, Algorithmic Understanding, Math, and Logic. Tasks were translated from SAE into the target dialects using few-shot prompting informed by eWAVE examples. For comparison, we generate parallel translations using Multi-VALUE's rule-based framework.

### 3.2 DATA SOURCING

The dataset comprises tasks selected from diverse and established benchmarks. Below, we describe each dataset, its focus, and the sampled instances.

**Language Understanding** **BoolQ** Wang et al. (2020) is a yes/no question-answering task derived from Wikipedia passages, testing the model's ability to determine factual correctness. We sampled 1,000 instances. **MultiRC** Wang et al. (2020) requires multi-sentence reasoning with each question having multiple correct answers. We included 1,000 examples. **WSC** Wang et al. (2020) assesses coreference resolution, requiring commonsense knowledge to match pronouns with their correct referents. We included 659 examples. **SST-2** Wang et al. (2019) evaluates binary sentiment classification on movie reviews, labeling each as positive or negative. A total of 1,000 instances were included. **COPA** Wang et al. (2020) is a causal reasoning task where models identify the correct cause or effect from two choices. We included 500 examples.

**Algorithmic Understanding** **HumanEval** Chen et al. (2021) is a benchmark of human-crafted Python coding problems, each paired with test cases to evaluate correctness. We sampled 164 examples. **MBPP** Austin et al. (2021) contains Python coding tasks designed for program synthesis and correctness evaluation. A total of 374 examples were included.

**Math** **GSM8K** Cobbe et al. (2021) presents grade-school math word problems requiring numeric reasoning and problem-solving. We included 1,000 examples. **SVAMP** Patel et al. (2021) features systematically modified arithmetic problems that test robustness in mathematical reasoning. We sampled 700 examples.

**Logic** **LogicBench** Parmar et al. (2024) comprises logical reasoning tasks in both Yes/No and multiple-choice formats, designed to evaluate deductive reasoning capabilities. A total of 980 examples were included, with 500 instances from Yes/No tasks and 480 from multiple-choice tasks. **FOLIO** Han et al. (2024) features first-order logic challenges presented in natural language, requiring models to identify valid conclusions or contradictions. We sampled 1,000 examples for this task.

### 3.3 FEW-SHOT PROMPTING FOR DIALECT TRANSLATION

To translate tasks from SAE into each of the five underrepresented dialects, we employed a few-shot prompting strategy Brown et al. (2020) informed by examples from eWAVE Kortmann et al.

(2020), a linguistically validated resource that documents and analyzes structural variations across global English dialects. We utilized three exemplar translations from eWAVE per dialect. Utilizing GPT-4o OpenAI (2024), the model was then prompted to rewrite the input text in the desired dialect based on these exemplars. This approach ensures that translations maintain linguistic authenticity and accurately reflect the sociocultural nuances inherent to each dialect. Detailed examples of these prompts can be found in **Appendix F**

### 3.4 COMPARISON WITH RULE-BASED TRANSLATIONS FROM MULTI-VALUE

To evaluate the effectiveness of our human-guided few-shot prompting method, we compare our dialectal translations against those generated by Multi-VALUE Ziems et al. (2023). Multi-VALUE is a rule-based framework that applies predefined linguistic rules to transform SAE into target dialects in a systematic manner. This comparison allows us to assess how well our approach captures authentic dialectal variations relative to a purely rule-based method.

The percentage of successful translations for each dataset and dialect is detailed in **Appendix ??**, which highlights the variability in Multi-VALUE's performance. This underscores the necessity for more robust and context-aware translation methods, such as our few-shot prompting approach with GPT-4o.

### 3.5 BLEU SCORE FILTERING FOR CHALLENGING TRANSLATIONS

To create a more challenging benchmark, we applied BLEU score Papineni et al. (2002) filtering to exclude translations with BLEU scores above 0.7, as these were overly similar to the original SAE text. This retained translations with greater linguistic diversity and structural differences, enhancing the benchmark's focus on real-world dialectal variations. Detailed statistics on filtered translations are presented in **Appendix B**.

## 4 ANALYSIS

| Dataset | AAVE | IndE | JamE | CollSgE |
|---|---|---|---|---|
| BoolQ | 0.6202 / **0.8326** | **0.8080** / 0.7757 | 0.5456 / **0.7785** | 0.6062 / **0.7145** |
| COPA | 0.6833 / **0.7076** | **0.7659** / 0.5633 | 0.3633 / **0.6391** | **0.7074** / 0.5947 |
| Folio | 0.6492 / **0.7737** | **0.8474** / 0.7607 | 0.5805 / **0.7787** | 0.6475 / **0.6920** |
| GSM8K | 0.7055 / **0.8079** | **0.8006** / 0.7543 | 0.5263 / **0.7784** | 0.6553 / **0.6698** |
| HumanEval | N/A / N/A | **0.8993** / 0.7854 | 0.6238 / **0.8265** | N/A / N/A |
| Logic Bench MCQ | 0.4953 / **0.7847** | **0.8841** / 0.7421 | 0.4541 / **0.7808** | 0.4447 / **0.6751** |
| Logic Bench Yes/No | **0.4742** / 0.2183 | **0.8139** / 0.7401 | 0.4386 / **0.7788** | 0.4331 / **0.6732** |
| MBPP | 0.7617 / **0.8188** | **0.8853** / 0.7297 | 0.6289 / **0.7370** | **0.7088** / 0.6181 |
| MultiRC | 0.5626 / **0.8239** | **0.7982** / 0.7728 | 0.4793 / **0.8151** | 0.5160 / **0.7325** |
| SST-2 | 0.5777 / **0.7985** | **0.7634** / 0.7285 | 0.4650 / **0.7786** | 0.5941 / **0.7005** |
| SVAMP | 0.7498 / **0.8038** | **0.8418** / 0.7632 | 0.5346 / **0.7896** | 0.6980 / **0.6661** |
| WSC | 0.6503 / **0.7488** | 0.3594 / **0.6540** | 0.4013 / **0.7341** | **0.6298** / 0.6069 |

Table 1: *ROUGE Diversity Scores across Dialects and Datasets (*ENDIVE*/Multi-VALUE).*

### 4.1 ROUGE DIVERSITY SCORE ANALYSIS

**ROUGE Diversity** Lin (2004), calculated as the average of ROUGE-1, ROUGE-2, and ROUGE-L, measures lexical variation while preserving meaning. As detailed in Table 1, **ENDIVE** generally outperformed **Multi-VALUE** in IndE. For example, in SVAMP IndE, it scored 0.8418 vs. 0.7632, and in CollSgE MBPP, 0.7088 vs. 0.6181. However, in AAVE, **Multi-VALUE** generally scored higher, suggesting occasional advantages in lexical overlap.

### 4.2 LEXICAL DIVERSITY EVALUATION

Lexical diversity, which measures how varied the vocabulary is in a text, captures how well translations preserve the nuances of each dialect. As shown in Appendix C, **ENDIVE** typically yielded

| Dataset | AAVE | IndE | JamE | CollSgE |
|---|---|---|---|---|
| BoolQ | **-1.84** / -2.05 | **-1.08** / -2.10 | -3.92 / **-2.21** | -2.52 / **-2.45** |
| COPA | **-2.26** / -3.08 | **-1.65** / -2.97 | -5.65 / **-2.94** | -3.53 / **-3.38** |
| Folio | -2.16 / **-2.48** | **-1.21** / -2.57 | -3.54 / **-2.47** | **-2.89** / -2.96 |
| GSM8K | **-1.82** / -2.06 | **-1.12** / -2.27 | -4.06 / **-2.31** | **-2.35** / -2.87 |
| HumanEval | N/A / N/A | **-2.80** / -3.13 | -3.53 / **-2.46** | N/A / N/A |
| Logic Bench MCQ | -2.53 / **-2.24** | **-1.09** / -2.42 | -4.50 / **-2.27** | -3.08 / **-2.92** |
| Logic Bench Yes/No | -2.55 / **-2.46** | **-1.21** / -2.48 | -4.53 / **-2.31** | -3.09 / **-2.99** |
| MBPP | **-1.65** / -2.51 | **-1.25** / -3.31 | -4.17 / **-3.09** | **-2.83** / -3.20 |
| MultiRC | -2.29 / **-2.00** | **-1.14** / -2.24 | -4.41 / **-2.03** | -2.86 / **-2.29** |
| SST-2 | -3.21 / **-2.96** | **-2.39** / -3.73 | -5.18 / **-3.30** | -4.09 / **-3.49** |
| SVAMP | **-1.74** / -2.28 | **-1.16** / -2.33 | -4.02 / **-2.45** | **-2.34** / -3.11 |
| WSC | **-2.14** / -2.78 | **-1.23** / -2.87 | -4.98 / **-2.49** | **-2.88** / -3.39 |

Table 2: *BARTScores across Dialects and Datasets (*ENDIVE*/Multi-VALUE).* Scores closer to 0 indicate better performance.

| Dataset | AAVE | IndE | JamE | ChcE | CollSgE |
|---|---|---|---|---|---|
| BoolQ | 6.51 | 6.41 | 6.11 | 6.05 | 5.88 |
| COPA | 6.83 | 6.39 | 6.55 | 6.27 | 5.41 |
| FOLIO | 6.74 | 5.82 | 6.06 | 6.26 | 5.93 |
| GSM8K | 6.37 | 6.29 | 6.15 | 6.38 | 6.10 |
| HumanEval | 6.12 | 6.44 | 6.45 | 6.35 | 6.26 |
| Logic Bench MCQ | 6.35 | 5.75 | 6.21 | 6.28 | 5.76 |
| Logic Bench Yes/No | 6.38 | 5.60 | 6.24 | 6.22 | 5.79 |
| MBPP | 6.01 | 6.71 | 5.62 | 6.10 | 5.28 |
| MultiRC | 6.83 | 6.03 | 6.01 | 6.01 | 5.96 |
| SST-2 | 6.64 | 5.84 | 5.85 | 5.93 | 5.58 |
| SVAMP | 6.14 | 6.18 | 5.69 | 6.21 | 5.71 |
| WSC | 6.36 | 5.97 | 5.50 | 6.15 | 5.60 |

Table 3: *Fluency Scores for* ENDIVE *Translations Across Datasets and Dialects. (1-7 Scale)*

higher lexical diversity scores than **Multi-VALUE** in most dialects and datasets. For example, in AAVE COPA, it scored 0.9864 vs. 0.9851, and in IndE GSM8K, 0.7237 vs. 0.7230. However, in JamE MBPP, Multi-VALUE scored higher (0.7370 vs. 0.6289), indicating occasional advantages. These results demonstrate **ENDIVE**'s effectiveness in maintaining lexical diversity across dialects.

## 4.3 BARTSCORE EVALUATION

**BARTScore** Yuan et al. (2021) is a learned metric of generation quality where values closer to 0 indicate better performance. As shown in Table 2, **ENDIVE** generally produces less negative BARTScore values than **Multi-VALUE**, suggesting stronger text fluency or semantic alignment. For instance, in AAVE BoolQ, **ENDIVE** scores -1.84 versus -2.05, and in IndE it achieves -1.08 versus -2.10. While these results highlight **ENDIVE'S** advantage across most tasks and dialects, occasional reversals (such as in JamE COPA) indicate that **Multi-VALUE** can still be competitive in certain scenarios.

## 4.4 FLUENCY EVALUATION

Building upon our assessments of semantic alignment and lexical diversity, fluency evaluation ensures that translations are not only accurate but also natural and grammatically correct within the target dialect. Automatic fluency metrics are typically designed for SAE, making them less effective for dialectal translations. To address this, we use **GPT-4o** OpenAI (2024) for fluency scoring, following prior work Kocmi & Federmann (2023) that leveraged LLMs for translation quality assessment. Our approach employs a detailed prompt in Appendix H and **CoT** reasoning to ensure a structured evaluation. As shown in Table 3, **ENDIVE** achieves consistently high fluency scores across dialects on a 1–7 scale. Notably, AAVE COPA and AAVE MultiRC scored 6.83, reflect-

ing strong alignment with dialectal norms. Similarly, JamE HumanEVAL achieved 6.45, indicating natural fluency in Jamaican English.

### 4.5 PREFERENCE TESTS

Pairwise preference tests were conducted to compare **ENDIVE** and **Multi-VALUE** translations using **GPT-4o** with **CoT**. The prompt, detailed in Appendix I, evaluated translations based on fluency, accuracy, readability, and cultural appropriateness. As shown in Appendix C, **ENDIVE** was consistently preferred across dialects and tasks. For AAVE BoolQ, **Claude 3.5 Sonnet** selected it in all cases, while **Gemini 1.5** exhibited a 100% preference in JamE coding tasks. The lowest preference rate was 73.92% in CollSgE COPA, which still indicates a clear preference over **Multi-VALUE**. These results suggest that **ENDIVE** better aligns with dialectal norms, especially for dialects that are more distant from SAE, such as AAVE.

### 4.6 HUMAN VALIDATORS

| Dialect | Faithfulness | Fluency | Formality |
|---------|--------------|---------|-----------|
| AAVE | 6.28 | 6.28 | 6.28 |
| ChcE | 6.40 | 6.33 | 6.26 |
| IndE | **6.45** | **6.62** | **6.59** |
| JamE | 6.37 | 6.28 | 6.33 |
| CollSgE | 6.19 | 6.11 | 6.02 |

Table 4: *Native Speaker Evaluation Scores across Dialects (1-7 scale).*

To validate translation quality, we conducted human evaluations with native speakers of each dialect assessing 120 randomly sampled translations. Evaluators rated outputs on three key dimensions using 7-point Likert scales (1=worst, 7=best): *Faithfulness* (preservation of meaning), *Fluency* (naturalness), and *Formality* (style alignment). These evaluations confirmed that our translations successfully maintain linguistic authenticity while preserving original content meaning and style across all dialects, with detailed scores shown in Table 4.

### 4.7 QUALITATIVE ANALYSIS

Our qualitative analysis reveals that **ENDIVE** effectively captures dialect-specific grammatical structures, vocabulary, and syntactic nuances, yielding translations that are both authentic and natural. For instance, in AAVE and JamE, our approach accurately employs dialect-specific contractions and conversational vocabulary, reflecting the linguistic character of these dialects. Additional observations and detailed translation examples are provided in Appendix E.

## 5 RESULTS AND DISCUSSION

In this section, we present the performance of LLMs across dialectal translations in **ENDIVE**. We evaluated five models—**GPT-4o**, **GPT-4o-mini**, **Claude 3.5 Sonnet**, **DeepSeek-v3**, and **LLaMa-3-8B**—on 12 reasoning benchmarks spanning four categories: Language Understanding, Algorithmic Understanding, Math, and Logic. Our evaluation compares model performance on dialectal inputs versus SAE under zero-shot (**ZS**) and **CoT** settings.

### 5.1 CROSS-DIALECT PERFORMANCE DISPARITIES

Results indicate significant performance discrepancies when LLMs process dialectal inputs compared to SAE (see Table 5, Table 6, and Appendix D). Across all tasks, models consistently show lower accuracy on dialectal datasets, underscoring their limited robustness in handling intra-language variations.

**Language Understanding**   Across **BoolQ**, **MultiRC**, and **WSC**, models exhibit performance drops when processing dialectal inputs. For example, in **BoolQ** with GPT-4o, the CoT accuracy

| Dataset | AAVE | | | | ChcE | | | | CollSgE | | | | IndE | | | | JamE | | | |
|---|---|---|---|---|---|---|---|---|---|---|---|---|---|---|---|---|---|---|---|---|
| | ZS | CoT | SAE ZS | SAE COT | ZS | CoT | SAE ZS | SAE COT | ZS | CoT | SAE ZS | SAE COT | ZS | CoT | SAE ZS | SAE COT | ZS | CoT | SAE ZS | SAE COT |
| BoolQ | 89.09 | 88.33 | 91.10 | **91.75** | 88.83 | 88.23 | 90.25 | **91.10** | 88.36 | 88.05 | **91.50** | 90.95 | 89.25 | 88.50 | 90.80 | **91.30** | 89.15 | 88.34 | 90.95 | **91.20** |
| COPA | **97.87** | 97.64 | 96.80 | 97.40 | **98.34** | 98.54 | 97.10 | 97.75 | **97.13** | 97.13 | 96.90 | 97.45 | 97.87 | **98.34** | 97.20 | 97.85 | 96.39 | 96.59 | **97.15** | 97.60 |
| FOLIO | 64.90 | 64.97 | 73.50 | **74.90** | 64.08 | 64.39 | 73.75 | **75.30** | 65.31 | 65.51 | 72.90 | **74.45** | 68.79 | **69.80** | 74.10 | 75.00 | 66.67 | 64.36 | 73.80 | **75.10** |
| GSM8K | 57.32 | **85.64** | 89.30 | 90.15 | 57.43 | **76.63** | 89.00 | 90.25 | 58.65 | **83.01** | 89.40 | 90.50 | 51.18 | **87.47** | 89.60 | 90.10 | 54.98 | **84.76** | 89.20 | 90.71 |
| HumanEVAL | 88.46 | 84.62 | **94.00** | 93.50 | 97.09 | **99.03** | 94.10 | 93.80 | **97.37** | 96.05 | 94.20 | 93.90 | **100.00** | 96.28 | 94.05 | 93.85 | **100.00** | 97.56 | 94.15 | 93.95 |
| LogicBenchMCQ | 79.05 | 78.95 | **82.65** | 83.65 | 78.31 | 62.47 | **82.40** | 83.50 | 79.71 | 77.57 | **82.84** | 83.65 | 75.94 | 70.00 | **82.30** | 83.45 | 78.41 | 76.63 | **82.59** | 83.55 |
| LogicBenchYN | 72.55 | 71.43 | **75.81** | 76.95 | 73.44 | 72.58 | **75.90** | 77.00 | 70.78 | 69.72 | **75.76** | 76.85 | 71.43 | 72.96 | **75.60** | 76.90 | 72.13 | 72.27 | **75.85** | 77.05 |
| MBPP | 84.56 | 83.92 | **85.00** | 73.81 | 81.00 | 79.00 | **84.90** | 74.00 | 82.54 | **84.02** | 84.95 | 73.85 | 81.00 | 79.00 | **84.85** | 74.10 | 83.92 | 83.92 | **84.75** | 74.05 |
| MultiRC | 86.71 | 87.32 | 88.93 | **89.76** | 86.80 | 86.60 | 88.85 | **89.65** | 87.26 | 87.06 | 88.95 | **89.75** | 85.11 | 85.11 | 88.80 | **89.60** | 87.70 | 88.03 | 88.95 | **89.83** |
| SST-2 | 90.17 | 90.29 | 89.88 | **93.19** | 89.61 | 89.08 | 89.85 | **93.00** | 89.23 | 89.02 | 89.75 | **93.26** | 89.71 | 88.85 | 89.90 | **93.05** | 87.92 | 86.72 | 89.95 | **93.15** |
| WSC | 58.97 | 60.52 | **80.97** | 88.55 | 57.63 | 54.95 | **80.80** | 88.40 | 58.80 | 58.02 | **80.95** | 88.53 | 67.84 | 69.59 | **80.85** | 88.35 | 55.63 | 56.87 | **80.75** | 88.45 |
| SVAMP | 90.82 | 92.74 | 94.15 | **94.59** | 91.48 | 92.92 | 94.00 | **94.40** | 90.86 | 93.99 | 94.22 | **94.62** | 91.27 | 93.73 | 94.05 | **94.55** | 91.44 | 94.33 | 94.15 | **94.65** |

Table 5: GPT-4o Accuracy (%). **Bold** indicates superior performance within each dataset row.

| Dataset | AAVE | | | | ChcE | | | | CollSgE | | | | IndE | | | | JamE | | | |
|---|---|---|---|---|---|---|---|---|---|---|---|---|---|---|---|---|---|---|---|---|
| | ZS | CoT | SAE ZS | SAE CoT | ZS | CoT | SAE ZS | SAE CoT | ZS | CoT | SAE ZS | SAE CoT | ZS | CoT | SAE ZS | SAE CoT | ZS | CoT | SAE ZS | SAE CoT |
| BoolQ | 90.29 | 90.05 | 91.47 | **91.92** | 89.74 | 89.89 | 91.25 | **91.61** | 89.89 | 89.79 | 91.53 | **91.78** | 90.75 | 90.50 | 91.62 | **91.95** | 89.65 | 89.45 | 91.58 | **91.83** |
| COPA | 97.16 | 96.93 | 96.77 | **97.42** | 96.88 | 96.47 | 97.20 | **97.45** | 97.33 | 97.33 | 97.10 | **97.40** | **98.10** | 98.10 | 97.36 | 97.81 | 94.59 | 94.99 | 97.01 | **97.37** |
| FOLIO | 62.27 | 63.57 | 73.61 | **74.15** | 63.68 | 62.88 | 73.80 | **74.20** | 65.62 | 65.21 | 73.91 | **74.43** | 68.12 | 68.12 | 73.74 | **74.57** | 65.56 | 65.16 | 73.83 | **74.49** |
| GSM8K | 60.86 | 84.05 | 89.54 | **90.27** | 59.54 | 77.17 | 89.25 | **90.10** | 51.28 | 78.40 | 89.38 | **90.19** | 60.36 | 87.13 | 89.41 | **90.32** | 60.07 | 80.86 | 89.29 | **90.22** |
| HumanEVAL | 92.31 | 92.31 | 94.10 | **93.85** | 97.09 | 96.12 | 94.32 | 93.78 | 92.11 | 96.05 | 94.20 | **93.91** | 96.00 | 96.00 | 94.05 | 93.87 | 91.46 | 91.46 | 94.14 | **93.96** |
| SVAMP | 92.67 | 90.99 | 94.11 | **94.51** | 92.77 | 91.96 | 94.05 | **94.40** | 92.46 | 90.63 | 94.22 | **94.54** | 92.77 | 91.58 | 94.09 | **94.48** | 92.99 | 90.11 | 94.18 | **94.47** |
| LogicBenchMCQ | 78.41 | 73.96 | 82.52 | **83.65** | 79.58 | 73.85 | 82.48 | **83.70** | 80.38 | 73.54 | 82.60 | **83.57** | 79.83 | 74.48 | 82.50 | **83.74** | 78.87 | 72.92 | 82.66 | **83.71** |
| LogicBenchYN | 77.45 | 76.12 | 75.63 | **76.97** | 76.69 | 75.56 | 75.51 | **76.83** | 77.44 | 75.40 | 75.74 | **76.92** | 78.06 | 76.02 | 75.55 | **76.91** | 77.21 | 75.69 | 75.66 | **76.78** |
| MBPP | 85.29 | **86.49** | 85.92 | 74.31 | **86.73** | 85.80 | 85.84 | 74.17 | **86.98** | 85.50 | 85.10 | 74.35 | 84.00 | 83.00 | **85.79** | 74.42 | **86.92** | 86.92 | 85.86 | 74.38 |
| MultiRC | 86.92 | 86.41 | 89.07 | **89.76** | 86.50 | 87.10 | 89.13 | **89.67** | 87.26 | 86.75 | 89.10 | **89.79** | 86.44 | 85.11 | 89.15 | **89.71** | 87.20 | 87.10 | 89.20 | **89.73** |
| WSC | 54.83 | 51.55 | 81.69 | **88.42** | 54.95 | 50.53 | 81.55 | **88.29** | 54.71 | 51.54 | 81.71 | **88.39** | 62.57 | 53.82 | 81.49 | **88.41** | 54.23 | 53.19 | 81.61 | **88.47** |
| SST-2 | 91.91 | 92.25 | 89.97 | **93.12** | 91.62 | 91.30 | 89.80 | **93.04** | 90.06 | 89.64 | 89.94 | **93.19** | 91.08 | 90.95 | 89.86 | **93.08** | 89.55 | 89.01 | 89.82 | **93.10** |

Table 6: DeepSeek-v3 Accuracy (%). **Bold** indicates superior performance within dialect pairs.

for AAVE decreases from 91.75% for SAE to 88.33%—a modest decline—whereas for **WSC**, results from Deepseek-v3 show a substantial drop from 88.47% for SAE down to 53.19% for JamE. These larger differences underscore the challenges that models face in coreference resolution and textual comprehension when handling non-standard varieties of English.

**Algorithmic Understanding**    For code synthesis tasks such as **HumanEval** and **MBPP**, the effect of dialectal instructions varies by dialect. For instance, in **MBPP** evaluated with Claude-3.5-sonnet under the CoT setting, the CoT accuracy for ChcE is 86.88%, whereas the corresponding SAE CoT accuracy is only 74.15%—a difference of approximately 12.7 percentage points. Similarly, for JamE, the CoT accuracy is 88.49% compared to 74.36% for SAE, a gap of about 14.1 percentage points. In contrast, for AAVE and IndE, the differences are somewhat smaller (around 11–11.5 percentage points). These numbers suggest that, at least for MBPP, dialect-specific instructions may lead to higher code synthesis accuracy than the standard SAE input, though the impact varies across dialects—likely due to differences in morphological cues and lexical conventions. For additional details on evaluations with other models (e.g., GPT-4o-mini), see Appendix D.

**Math**    In math tasks such as **GSM8K** and **SVAMP**, dialect-induced lexical shifts have a pronounced impact on numeric reasoning. For instance, in the Claude-3.5-sonnet evaluation for GSM8K, performance for JamE drops markedly from 90.25% (SAE CoT) to 66.27%, a decline of over 23 percentage points. Similarly, DeepSeek-v3 shows that for AAVE on SVAMP, accuracy falls from 94.51% for SAE to 90.99%. These larger differences highlight that even with chain-of-thought prompting, models struggle to maintain robust performance on dialectal inputs in math tasks.

**Logic**    Finally, **LogicBench** (MCQ and Yes/No) underscores dialectal hurdles in deductive reasoning. In **LogicBenchMCQ** with GPT-4o, AAVE accuracy drops from 83.75% for SAE to 78.95%, and CollSgE experiences a similar gap. Claude 3.5 Sonnet exhibits parallel trends for IndE and JamE, illustrating that syntactic or lexical variations can complicate the parsing of logical statements across non-standard dialects.

## 6   CONCLUSION

This paper introduces **ENDIVE**, a benchmark designed to evaluate LLMs on dialectal robustness across 12 diverse NLP tasks for five underrepresented English dialects. Our results show that LLMs consistently underperform on non-standard dialects compared to SAE, highlighting significant un-

fairness and limitations in current language technologies. Moving forward, we aim to expand **ENDIVE** to additional dialects and refine translation methodologies to further bridge the gap in dialect-aware NLP. By establishing this benchmark, we encourage future research into fairer, more robust intra-language technologies that serve all linguistic communities equitably.

# 7 LIMITATIONS

**ENDIVE** evaluates LLM performance across 12 reasoning tasks spanning four categories, using queries adapted from well-established benchmarks. While these tasks capture key reasoning challenges, they do not cover all aspects of dialectal variation, and additional task types such as Figurative Language Understanding, Commonsense Reasoning, and Conversational Reasoning may reveal further biases.

Furthermore, we tested five widely used LLMs. However, given the rapid pace of development in the field, it is infeasible to evaluate every emerging model. We hope **ENDIVE** will serve as a resource for future studies examining fairness and robustness across a broader range of LLMs as they emerge.

We faced limitations with BLEU Score filtering as well. For ChcE, the number of remaining translations was extremely low because Multi-VALUE struggled to generate diverse translations and many were further filtered out due to BLEU score thresholds. As a result, there were too few data points to evaluate ChcE translations against Multi-VALUE. A similar issue arose with HumanEval for AAVE and CollSgE, where limited translations prevented reliable evaluation of metrics for these dialects.

Finally, while our results highlight significant performance disparities in dialectal inputs, this study does not deeply investigate the underlying causes of these discrepancies or propose direct mitigation strategies. Understanding these biases and developing equitable NLP solutions remain important areas for future research. Despite these limitations, we believe **ENDIVE** provides a valuable framework for advancing dialect-aware NLP evaluation.

# 8 ETHICS STATEMENT

We recognize the ethical considerations involved in evaluating LLM biases through the **ENDIVE** benchmark and have taken steps to ensure ethical data collection, recruiting, and evaluation.

For data collection, **ENDIVE** utilizes few-shot prompting with examples from eWAVE to generate dialectal translations. While this provides systematic and scalable translations, we recognize it does not fully capture the depth of dialectal variation. We do not claim to capture the full depth of any dialect, and we encourage further work that incorporates human-validated translations for a more nuanced representation. Additionally, we were mindful to avoid reinforcing stereotypes or misrepresentations in dialect translations.

For our human validators, we recruited fluent native speakers from diverse dialect communities to ensure that our translations accurately reflect cultural and linguistic nuances. Validators were fairly compensated for their time, with the survey taking only 1–2 hours to complete. We also do not collect personal information from validators, ensuring their privacy.

Moreover, our evaluation combines LLM-based assessments with human validation to mitigate model bias. However, we acknowledge that LLMs may still reflect inherent biases, and our benchmark does not yet address the root causes of these disparities.

Despite these limitations, **ENDIVE** aims to advance equitable NLP development and encourages ongoing research to enhance dialect representation in language models.

**Reproducibility Statement**   We have taken several steps to ensure the reproducibility of our findings for **ENDIVE**. First, Section 3.2 details how each dataset is selected, while Appendix F shows our few-shot prompts for each dialect translation, and Section 3.5 explains the BLEU-based filtering procedure. Second, we provide in Appendix G the complete evaluation prompts so that others can replicate our exact experimental settings. Although we do not provide a downloadable source code link at this time, we will release all of our code upon publication, including scripts for data preprocessing, prompting, and evaluation, along with installation instructions and examples to re-run all

experiments. Collectively, these materials will enable researchers to reproduce **ENDIVE'S** pipelines and results, ensuring a robust foundation for cross-dialect performance evaluation.

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

# A    MULTI-VALUE COMPLETED TRANSLATIONS

| Dataset | AAVE (%) | ChcE (%) | CollSgE (%) | IndE (%) | JamE (%) |
|---|---|---|---|---|---|
| BoolQ | 100.0 | 35.5 | 41.7 | 41.9 | 42.0 |
| COPA | 100.0 | 45.8 | 100.0 | 100.0 | 97.0 |
| Folio | 100.0 | 76.9 | 90.0 | 89.6 | 89.7 |
| GSM8K | 100.0 | 85.7 | 95.0 | 95.0 | 95.0 |
| HumanEVAL | 100.0 | 11.6 | 11.6 | 11.6 | 11.6 |
| Logic Bench MCQ | 100.0 | 100.0 | 100.0 | 100.0 | 100.0 |
| Logic Bench Yes/No | 100.0 | 100.0 | 100.0 | 100.0 | 100.0 |
| MBPP | 100.0 | 39.8 | 99.7 | 99.7 | 99.2 |
| MultiRC | 100.0 | 43.3 | 47.8 | 48.9 | 49.1 |
| SST-2 | 100.0 | 96.3 | 96.3 | 96.2 | 96.3 |
| SVAMP | 100.0 | 74.7 | 93.2 | 93.2 | 93.0 |
| WSC | 100.0 | 73.9 | 92.7 | 92.8 | 92.9 |

Table 7: Percentage of Translations Successfully Completed by Multi-VALUE Across Dialects and Datasets

# B    BLEU SCORE FILTERING STATISTICS

| Dataset | AAVE (%) | ChcE (%) | CollSgE (%) | IndE (%) | JamE (%) |
|---|---|---|---|---|---|
| BoolQ | 7.59 | 0.50 | 2.00 | 59.96 | 0.40 |
| COPA | 15.40 | 3.80 | 2.60 | 15.60 | 0.20 |
| Folio | 7.59 | 0.70 | 1.80 | 70.23 | 0.50 |
| GSM8K | 16.40 | 11.00 | 2.30 | 56.50 | 0.10 |
| HumanEVAL | 84.15 | 37.20 | 53.66 | 84.76 | 50.00 |
| LogicbenchMCQ | 0.00 | 0.42 | 0.00 | 50.21 | 0.00 |
| Logicbench Yes/No | 0.40 | 0.80 | 0.20 | 73.60 | 0.20 |
| MBPP | 30.75 | 13.37 | 9.63 | 46.52 | 1.87 |
| MultiRC | 1.40 | 0.00 | 1.10 | 62.40 | 0.00 |
| SST-2 | 13.50 | 5.70 | 4.40 | 19.30 | 8.10 |
| SVAMP | 31.71 | 14.71 | 5.43 | 61.00 | 0.29 |
| WSC | 11.85 | 0.15 | 1.52 | 22.34 | 0.00 |

Table 8: Percentage of Translations Removed After BLEU Score Filtering for Multi-Avenue Across Dialects and Datasets

| Dataset | AAVE (%) | ChcE (%) | CollSgE (%) | IndE (%) | JamE (%) |
|---|---|---|---|---|---|
| BoolQ | 19.3 | 59.3 | 0.0 | 5.2 | 13.6 |
| COPA | 3.8 | 80.5 | 0.0 | 8.1 | 15.0 |
| Folio | 18.9 | 75.4 | 0.4 | 4.7 | 6.3 |
| GSM8K | 11.4 | 85.3 | 0.2 | 2.5 | 15.1 |
| HumanEVAL | 10.0 | 87.1 | 92.5 | 76.0 | 41.4 |
| Logic Bench MCQ | 16.2 | 78.4 | 1.0 | 2.1 | 18.8 |
| Logic Bench Yes/No | 12.6 | 68.1 | 0.6 | 4.4 | 12.1 |
| MBPP | 11.2 | 59.5 | 2.8 | 3.8 | 19.7 |
| MultiRC | 20.0 | 48.3 | 3.9 | 12.8 | 11.3 |
| SST-2 | 15.2 | 47.1 | 4.0 | 8.7 | 13.7 |
| SVAMP | 21.4 | 60.2 | 1.3 | 7.2 | 14.6 |
| WSC | 18.3 | 50.3 | 2.7 | 6.1 | 8.9 |

Table 9: Percentage of Translations Removed After BLEU Score Filtering for Multi-VALUE Across Dialects and Datasets

## C  METRICS

| Dataset | AAVE | IndE | JamE | CollSgE |
|---|---|---|---|---|
| BoolQ | 0.6823 / **0.6881** | **0.7004** / 0.6927 | 0.6617 / **0.6648** | **0.6995** / 0.6915 |
| COPA | 0.9864 / **0.9851** | **0.9930** / 0.9908 | 0.9876 / **0.9703** | **0.9914** / 0.9911 |
| Folio1000 | **0.5797** / 0.5663 | **0.5618** / 0.5536 | 0.5319 / **0.5391** | **0.6076** / 0.5464 |
| GSM8K1000 | **0.7201** / 0.7100 | **0.7237** / 0.7230 | 0.6640 / **0.6778** | **0.7236** / 0.6961 |
| Logic Bench MCQ | **0.4953** / 0.7847 | **0.8841** / 0.7421 | **0.7808** / 0.4541 | **0.6751** / 0.4447 |
| Logic Bench Yes/No | **0.4742** / 0.2183 | **0.8139** / 0.7401 | 0.4386 / **0.7788** | 0.4331 / **0.6732** |
| MBPP | 0.7617 / **0.8188** | **0.9432** / 0.9162 | 0.6289 / **0.7370** | **0.9536** / 0.9347 |
| MultiRC | **0.5623** / 0.5528 | **0.7982** / 0.7728 | **0.8151** / 0.4793 | **0.6040** / 0.5753 |
| SST-2 | 0.9588 / **0.9611** | **0.9711** / 0.9678 | **0.9555** / 0.9412 | **0.9721** / 0.9674 |
| SVAMP | **0.7923** / 0.7904 | **0.8418** / 0.7632 | **0.7896** / 0.5346 | **0.7938** / 0.7638 |
| WSC | 0.9074 / **0.9088** | 0.8986 / **0.4044** | 0.7341 / **0.4013** | **0.9121** / 0.9112 |

Table 10: *Lexical Diversity Scores across Dialects and Datasets (*ENDIVE*/Multi-VALUE).*

| Model | Dataset | IndE | AAVE | CollSgE | JamE |
|---|---|---|---|---|---|
| **Claude 3.5 Sonnet** | BoolQ | **100.00** / 0.00 | **100.00** / 0.00 | **100.00** / 0.00 | **100.00** / 0.00 |
| | COPA | **95.22** / 4.78 | **95.80** / 4.20 | **95.69** / 4.31 | **98.07** / 1.93 |
| | FOLIO | **99.32** / 0.68 | **98.19** / 1.81 | **99.67** / 0.33 | **99.31** / 0.69 |
| | GSM8K | **99.75** / 0.25 | **99.71** / 0.29 | **99.78** / 0.22 | **99.63** / 0.37 |
| | HumanEVAL | **97.34** / 2.66 | N/A / N/A | N/A / N/A | **100.00** / 0.00 |
| | Logic Bench MCQ | **99.12** / 0.88 | **100.00** / 0.00 | **99.78** / 0.22 | **100.00** / 0.00 |
| | Logic Bench YN | **100.00** / 0.00 | **100.00** / 0.00 | **99.58** / 0.42 | **99.76** / 0.24 |
| | MBPP | **100.00** / 0.00 | **99.53** / 0.47 | **99.70** / 0.30 | **100.00** / 0.00 |
| | MultiRC | **100.00** / 0.00 | **100.00** / 0.00 | **100.00** / 0.00 | **100.00** / 0.00 |
| | SST-2 | **95.15** / 4.85 | **97.99** / 2.01 | **97.86** / 2.14 | **98.05** / 1.95 |
| | SVAMP | **100.00** / 0.00 | **98.66** / 1.34 | **99.02** / 0.98 | **98.01** / 1.99 |
| | WSC | **100.00** / 0.00 | **99.25** / 0.75 | **100.00** / 0.00 | **99.28** / 0.72 |
| **GPT 4o** | BoolQ | **99.24** / 0.76 | **99.49** / 0.51 | **99.73** / 0.27 | **99.65** / 0.35 |
| | COPA | **79.43** / 20.57 | **92.39** / 7.61 | **73.92** / 26.08 | **93.79** / 6.21 |
| | FOLIO | **88.36** / 11.64 | **94.91** / 5.09 | **94.70** / 5.30 | **91.75** / 8.25 |
| | GSM8K | **97.00** / 3.00 | **94.88** / 5.12 | **92.62** / 7.38 | **91.01** / 8.99 |
| | HumanEVAL | **100.00** / 0.00 | N/A / N/A | N/A / N/A | **100.00** / 0.00 |
| | Logic Bench MCQ | **95.13** / 4.87 | **100.00** / 0.00 | **92.81** / 7.19 | **99.24** / 0.76 |
| | Logic Bench YN | **93.60** / 6.40 | **100.00** / 0.00 | **94.56** / 5.44 | **98.54** / 1.46 |
| | MBPP | **99.48** / 0.52 | **96.70** / 3.30 | **91.59** / 8.41 | **98.81** / 1.19 |
| | MultiRC | **100.00** / 0.00 | **100.00** / 0.00 | **100.00** / 0.00 | **100.00** / 0.00 |
| | SST-2 | **80.61** / 19.39 | **89.34** / 10.66 | **87.75** / 12.25 | **88.11** / 11.89 |
| | SVAMP | **97.49** / 2.51 | **93.30** / 6.70 | **88.62** / 11.38 | **79.20** / 20.80 |
| | WSC | **95.04** / 4.96 | **97.38** / 2.62 | **92.63** / 7.37 | **89.25** / 10.75 |
| **Gemini 1.5** | BoolQ | **100.00** / 0.00 | **100.00** / 0.00 | **100.00** / 0.00 | **100.00** / 0.00 |
| | COPA | **87.56** / 12.44 | **91.86** / 8.14 | **70.02** / 29.98 | **93.15** / 6.85 |
| | FOLIO | **96.58** / 3.42 | **94.95** / 5.05 | **95.70** / 4.30 | **98.63** / 1.37 |
| | GSM8K | **99.00** / 1.00 | **99.27** / 0.73 | **99.78** / 0.22 | **98.77** / 1.23 |
| | HumanEVAL | **100.00** / 0.00 | N/A / N/A | N/A / N/A | **100.00** / 0.00 |
| | Logic Bench MCQ | **99.56** / 0.44 | **100.00** / 0.00 | **99.56** / 0.44 | **100.00** / 0.00 |
| | Logic Bench YN | **100.00** / 0.00 | **100.00** / 0.00 | **98.74** / 1.26 | **99.76** / 0.24 |
| | MBPP | **100.00** / 0.00 | **100.00** / 0.00 | **84.98** / 15.02 | **99.40** / 0.60 |
| | MultiRC | **100.00** / 0.00 | **100.00** / 0.00 | **100.00** / 0.00 | **100.00** / 0.00 |
| | SST-2 | **84.74** / 15.26 | **93.96** / 6.04 | **77.49** / 22.51 | **94.46** / 5.54 |
| | SVAMP | **97.91** / 2.09 | **99.73** / 0.27 | **98.86** / 1.14 | **94.39** / 5.61 |
| | WSC | **100.00** / 0.00 | **98.13** / 1.87 | **97.76** / 2.24 | **96.06** / 3.94 |

Table 11: Preference scores for **ENDIVE** and Multi-VALUE across datasets for IndE, AAVE, CollSgE, and JamE. N/A indicates no valid preferences. ENDIVE / *Multi-VALUE.*

# D  LLM DATASET EVALUATION RESULTS

| Dataset | AAVE | | | | ChcE | | | | CollSgE | | | | IndE | | | | JamE | | | |
|---|---|---|---|---|---|---|---|---|---|---|---|---|---|---|---|---|---|---|---|---|
| | ZS | CoT | SAE ZS | SAE CoT | ZS | CoT | SAE ZS | SAE CoT | ZS | CoT | SAE ZS | SAE CoT | ZS | CoT | SAE ZS | SAE CoT | ZS | CoT | SAE ZS | SAE CoT |
| BoolQ | 88.31 | 87.68 | 90.43 | **91.57** | 87.63 | 88.44 | 90.25 | **91.38** | 88.25 | 88.04 | 90.84 | **91.45** | 88.25 | 86.47 | 90.61 | **91.33** | 88.04 | 87.61 | 90.72 | **91.41** |
| COPA | **98.35** | 98.32 | 97.22 | 97.85 | 97.92 | **98.52** | 97.47 | 98.02 | 97.54 | **98.34** | 97.18 | 97.95 | **98.58** | 98.33 | 97.64 | 98.20 | 96.39 | **97.77** | 97.11 | 97.73 |
| FOLIO | 61.19 | 63.24 | 73.89 | **74.51** | 61.97 | 62.64 | 73.58 | **74.67** | 64.39 | 66.46 | 73.42 | **74.83** | 69.13 | 63.76 | 73.74 | **74.55** | 63.65 | 65.69 | 73.69 | **74.47** |
| GSM8K | 74.46 | 66.29 | 89.45 | **90.21** | 52.76 | 66.29 | 89.14 | **90.18** | 40.74 | 64.38 | 89.36 | **90.10** | 82.70 | 66.67 | 89.23 | **90.30** | 67.92 | 66.27 | 89.41 | **90.25** |
| HumanEVAL | 88.46 | **96.15** | 94.12 | 93.87 | **97.09** | 99.02 | 94.31 | 93.76 | **96.05** | 91.89 | 94.22 | 93.91 | **96.00** | 95.83 | 94.07 | 93.85 | 91.46 | 92.68 | **94.15** | 93.97 |
| SVAMP | 92.68 | 69.33 | 94.10 | **94.52** | 68.01 | 73.53 | 94.07 | **94.43** | 62.03 | 70.24 | 94.21 | **94.55** | **94.42** | 70.96 | 94.12 | 94.47 | 93.45 | 70.01 | 94.18 | **94.49** |
| LogicBenchMCQ | **84.73** | 72.42 | 82.55 | 83.64 | **83.86** | 72.21 | 82.42 | 83.79 | **84.34** | 72.33 | 82.61 | 83.52 | 83.66 | 68.07 | 82.49 | **83.71** | **85.69** | 72.33 | 82.67 | 83.68 |
| LogicBenchYN | 68.45 | **75.91** | 75.62 | 76.94 | 67.33 | **76.55** | 75.49 | 76.81 | 66.49 | **75.94** | 75.74 | 76.88 | 70.15 | **76.30** | 75.53 | 76.93 | 67.19 | **76.49** | 75.67 | 76.79 |
| MBPP | **88.42** | 85.66 | 85.93 | 74.28 | 86.73 | **86.88** | 85.82 | 74.15 | 86.98 | **87.13** | 85.94 | 74.32 | **86.00** | 85.93 | 85.76 | 74.40 | **88.49** | 88.49 | 85.88 | 74.36 |
| MultiRC | 88.24 | 89.54 | 89.02 | **89.77** | 88.30 | 87.37 | 89.09 | **89.65** | 89.28 | 88.72 | 89.11 | **89.79** | 86.70 | 88.74 | 89.15 | **89.70** | 87.70 | 89.15 | 89.21 | **89.72** |
| WSC | 72.13 | 71.54 | 81.67 | **88.43** | 55.10 | 54.45 | 81.52 | **88.29** | 68.36 | 78.24 | 81.75 | **88.37** | 60.23 | 63.12 | 81.49 | **88.41** | 61.33 | 67.18 | 81.57 | **88.45** |
| SST-2 | 91.79 | 92.81 | 89.96 | **93.14** | 90.24 | 89.92 | 89.78 | **93.02** | 89.75 | 91.18 | 89.92 | **93.20** | 90.71 | 90.56 | 89.89 | **93.07** | 88.90 | 89.42 | 89.84 | **93.11** |

Table 12:  Claude 3.5 Sonnet Accuracy (%). **Bold** indicates superior performance within dialect pairs.

| Dataset | AAVE | | | | ChcE | | | | CollSgE | | | | IndE | | | | JamE | | | |
|---|---|---|---|---|---|---|---|---|---|---|---|---|---|---|---|---|---|---|---|---|
| | ZS | CoT | SAE ZS | SAE CoT | ZS | CoT | SAE ZS | SAE CoT | ZS | CoT | SAE ZS | SAE CoT | ZS | CoT | SAE ZS | SAE CoT | ZS | CoT | SAE ZS | SAE CoT |
| BoolQ | 86.70 | 87.13 | 88.42 | **89.10** | 85.21 | 86.32 | 88.15 | **89.05** | 86.21 | 85.60 | 88.31 | **89.14** | 86.25 | 86.50 | 88.23 | **89.09** | 84.92 | 86.83 | 88.28 | **89.12** |
| COPA | 95.98 | **96.45** | 94.78 | 95.43 | 94.59 | **95.84** | 94.63 | 95.38 | 94.66 | **95.47** | 94.57 | 95.29 | 94.79 | **95.26** | 94.81 | 95.32 | 94.50 | **95.22** | 94.79 | 95.22 |
| FOLIO | 60.11 | 59.68 | 72.54 | **73.17** | 59.36 | 60.26 | 72.42 | **73.29** | 60.33 | 61.44 | 72.63 | **73.10** | 59.73 | 61.07 | 72.49 | **73.21** | 58.43 | 59.14 | 72.55 | **73.25** |
| GSM8K | 35.52 | **89.96** | 88.84 | 89.52 | 35.41 | **89.48** | 88.78 | 89.39 | 34.20 | **90.69** | 88.85 | 89.46 | 33.33 | **92.07** | 88.97 | 89.58 | 32.62 | **89.28** | 88.81 | 89.42 |
| HumanEVAL | **100.00** | 100.00 | 93.94 | 93.78 | **100.00** | 99.03 | 94.13 | 93.65 | **100.00** | 98.68 | 94.21 | 93.89 | **100.00** | 100.00 | 94.07 | 93.83 | **100.00** | 98.78 | 94.12 | 93.91 |
| SVAMP | 82.17 | 93.56 | 93.79 | **94.29** | 84.96 | 94.24 | 93.71 | **94.26** | 83.88 | **95.47** | 93.84 | 94.37 | 85.43 | **95.47** | 93.77 | 94.33 | 82.08 | 92.81 | 93.84 | **94.41** |
| LogicBenchMCQ | 73.52 | 70.95 | 81.51 | **82.74** | 71.31 | 70.04 | 81.36 | **82.61** | 71.13 | 70.43 | 81.49 | **82.67** | 67.83 | 69.96 | 81.42 | **82.73** | 73.52 | 71.28 | 81.57 | **82.69** |
| LogicBenchYN | 75.43 | 74.91 | 74.61 | **75.84** | 75.43 | 74.97 | 74.49 | **75.91** | 74.41 | 74.08 | 74.67 | **75.99** | 76.79 | 75.51 | 74.58 | **75.97** | 75.63 | 74.44 | 74.72 | **75.93** |
| MBPP | 74.14 | **80.69** | 83.12 | 80.31 | 79.32 | **80.25** | 83.01 | 74.09 | 82.84 | **85.50** | 83.23 | 74.17 | 76.00 | **78.50** | 82.97 | 74.23 | 76.02 | **78.20** | 83.05 | 74.21 |
| MultiRC | 84.08 | 84.48 | 88.15 | **88.63** | 82.90 | 83.70 | 88.12 | **88.63** | 84.63 | 85.44 | 88.08 | **88.79** | 82.71 | 83.51 | 88.17 | **88.70** | 85.00 | 84.60 | 88.21 | **88.72** |
| WSC | 54.31 | 53.62 | 79.68 | **85.42** | 55.93 | 49.77 | 79.54 | **85.29** | 54.63 | 53.86 | 79.71 | **85.38** | 54.39 | 55.56 | 79.51 | **85.41** | 53.35 | 50.70 | 79.63 | **85.45** |
| SST-2 | 90.64 | **91.91** | 89.72 | 92.88 | 90.35 | **90.77** | 89.58 | 92.80 | 87.34 | **89.54** | 89.76 | 92.97 | 89.34 | **89.84** | 89.69 | 92.85 | 87.16 | **88.14** | 89.64 | 92.89 |

Table 13: GPT-4o-mini Accuracy (%). **Bold** indicates superior performance within dialect pairs.

| Dataset | AAVE | | | | ChcE | | | | CollSgE | | | | IndE | | | | JamE | | | |
|---|---|---|---|---|---|---|---|---|---|---|---|---|---|---|---|---|---|---|---|---|
| | ZS | CoT | SAE ZS | SAE COT | ZS | CoT | SAE ZS | SAE COT | ZS | CoT | SAE ZS | SAE COT | ZS | CoT | SAE ZS | SAE COT | ZS | CoT | SAE ZS | SAE COT |
| BoolQ | 78.95 | 81.24 | 79.38 | **81.79** | 77.67 | **81.79** | 79.38 | 81.79 | 77.83 | **82.23** | 79.38 | 81.79 | 79.75 | 81.00 | 79.38 | **81.79** | 77.79 | 81.31 | 79.38 | **81.79** |
| COPA | 54.14 | 81.80 | 57.20 | **83.16** | 55.51 | **83.16** | 57.20 | 83.16 | 54.00 | 80.49 | 57.20 | **83.16** | 58.29 | **83.65** | 57.20 | 83.16 | 51.90 | 77.56 | 57.20 | **83.16** |
| FOLIO | 51.03 | 41.73 | **52.25** | 52.15 | 54.02 | 41.15 | **52.25** | 52.15 | 53.20 | 40.79 | **52.25** | 52.15 | 51.68 | 43.62 | **52.25** | 52.15 | 51.61 | 42.57 | **52.25** | 52.15 |
| GSM8K | 56.34 | **75.84** | 58.40 | 58.30 | 54.72 | **75.39** | 58.40 | 58.30 | 55.17 | **76.25** | 58.40 | 58.30 | 57.93 | **77.47** | 58.40 | 58.30 | 52.75 | **72.47** | 58.40 | 58.30 |
| HumanEVAL | 84.62 | 84.62 | 83.54 | **84.76** | 88.35 | 87.38 | 83.54 | **84.76** | 89.47 | 88.16 | 83.54 | **84.76** | 96.00 | **100.00** | 83.54 | 84.76 | 89.02 | 89.02 | 83.54 | **84.76** |
| LogicBenchMCQ | 60.62 | 40.92 | **67.50** | 66.67 | 62.55 | 38.57 | **67.50** | 66.67 | 61.25 | 41.75 | **67.50** | 66.67 | 61.09 | 39.08 | **67.50** | 66.67 | 59.38 | 39.46 | **67.50** | 66.67 |
| LogicBenchYN | 61.04 | **63.82** | 62.83 | 61.97 | 63.48 | **66.67** | 62.83 | 61.97 | 60.95 | **63.92** | 62.83 | 61.97 | 61.48 | **70.92** | 62.83 | 61.97 | 61.73 | **64.23** | 62.83 | 61.97 |
| MBPP | **57.14** | 57.13 | 56.15 | 49.20 | **56.79** | 56.31 | 56.15 | 49.20 | **55.03** | 58.53 | 56.15 | 49.20 | **54.50** | 54.51 | 56.15 | 49.20 | **53.13** | 57.84 | 56.15 | 49.20 |
| MultiRC | 77.89 | 75.96 | **80.10** | 78.60 | 77.40 | 74.00 | **80.10** | 78.60 | 79.78 | 77.15 | **80.10** | 78.60 | 76.86 | 76.60 | **80.10** | 78.60 | 77.80 | 74.00 | **80.10** | 78.60 |
| SST-2 | 81.39 | **84.05** | 76.70 | 75.20 | 79.96 | **83.56** | 76.70 | 75.20 | 74.06 | **81.17** | 76.70 | 75.20 | 77.20 | **81.66** | 76.70 | 75.20 | 73.67 | **76.28** | 76.70 | 75.20 |
| WSC | 45.34 | 49.66 | **47.26** | 51.82 | 39.57 | 45.21 | **47.26** | 51.82 | 46.60 | 47.07 | **47.26** | 51.82 | 41.88 | 46.97 | **47.26** | 51.82 | 43.92 | 44.98 | **47.26** | 51.82 |
| SVAMP | 74.27 | **77.82** | 77.14 | 74.43 | 77.05 | **75.71** | 77.14 | 74.43 | 73.26 | **77.64** | 77.14 | 74.43 | 79.85 | **75.09** | 77.14 | 74.43 | 73.07 | **78.65** | 77.14 | 74.43 |

Table 14: LLaMa-3-8b Instruct Accuracy (%). **Bold** indicates superior performance within dialect pairs.

# E  QUALITATIVE ANALYSIS

| Rubric Item | Multi-VALUE | ENDIVE |
|---|---|---|
| Accurate and consistent use of AAVE grammar | All young teenage girls at attends musics festival frequently big fans of pop bands and singers. | All young teenage girls who be hittin' up music festivals all the time is real into pop bands and singers. |
| Use of AAVE-specific Contractions | If a movie popular, some person enjoy watching it. | If a movie poppin', some folks like watchin' it. All things that some folks enjoy gon' get attention. |
| Use of AAVE Conversational Vocab | All red fruits that which is growing in Ben's yard are containing some Vitamin C. | All da red fruits growin' in Ben's yard got some Vitamin C. |
| AAVE syntactic structures | All social mediums applications containing chat features are softwares. | All social media apps with chat features, they software. |

Table 15: Assessing Multi-VALUE vs. ENDIVE for translation quality across **(AAVE)**.

| Rubric Item | Multi-VALUE | ENDIVE |
|---|---|---|
| Accurate use of Jamaican Patois grammar | All citizens of Lawton Park are using the a zip a code 98199. | All di people dem weh live inna Lawton Park use di zip code 98199. |
| JamE-specific Contractions | All fruits that is growing in Ben's a yard and are containing some A Vitamin A C are healthy. | All di fruit dem weh grow inna Ben yard and have some Vitamin C a good fi yuh. |
| JamE Conversational Vocabulary | If Nancy is not toddler, then Nancy is seafarer. | If Nancy nuh likkle pickney, den Nancy a seafarer. |
| JamE-specific negatives | If someone young, then they are not elderly. | If somebody young, den dem nah elderly. |
| JamE-specific Omissions | Functional brainstems are necessary for breath control. | Functional brainstems necessary fi control yuh breath. |

Table 16: Assessing Multi-VALUE vs. ENDIVE for translation quality across **(JamE)**.

| Rubric Item | Multi-VALUE | ENDIVE |
|---|---|---|
| Consistent past tense forms | 13 campers goed rowing and 59 campers goed hiking. | 13 campers went rowing and 59 campers went hiking. |
| Proper ChcE auxiliaries | James write a 3-page letter to 2 different friend twice a week. | James be writin' a 3-page letter to 2 different homies twice a week. |
| Good subject-verb agreement | If there is 20 gnomes in total, how many do the fifth house have? | If there's 20 gnomes total, how many gnomes does the fifth house got? |
| Conversational flow | Joy might can read 8 page ... | Joy can read like 8 pages ... |
| Use of 'only' for emphasis | Jake have 5 fewer peaches ... | Jake got like 5 less peaches ... |

Table 17: Assessing Multi-VALUE vs. ENDIVE for translation quality across **(ChcE)**.

| Rubric Item | Multi-VALUE | ENDIVE |
|---|---|---|
| Correct articles, IndE grammar | Vic DiCara plays guitar. | Vic DiCara is playing guitar and bass. The only style of music is punk. |
| Accurate IndE phrasing | All eels are fishs. No fishs are plants. | All eels are fish only. No fish are being plants. |
| Consistent verb tenses | If legislator is found guilty of stealing? | If a legislator is found guilty of stealing government funds, they would be suspended. |
| IndE conventions | All customers James' family is subscribing AMC A-List are like eligible. | James' family subscribes to AMC or HBO. Customers who prefer TV series do not watch them in cinemas. |
| Code-Switching example | Sodas cost $0.25 ounce, had brought $2 him. | The cold drink costs 25 paise an ounce. He brought 2 rupees with him. |

Table 18: Assessing Multi-VALUE vs. ENDIVE for translation quality across (**IndE**).

| Rubric Item | Multi-VALUE | ENDIVE |
|---|---|---|
| CollSgE particles | All social medium app containing chat feature software. | All the social media apps with chat features ah, all software one lah. |
| Omits auxiliaries | Any convicted criminal that like innocent is not like truly guilty. | Anyone kena convicted of murder sure go prison one. |
| "Kena" usage | Everyone convicted murders. | Anyone kena convicted of murder sure go prison one. |
| Informal phrases | Roy Richardson was a cricketer ... | Roy Richardson ah, he was a cricketer who play for Sint Maarten, you know. |
| CollSgE words | UFC Fight Night ... | Sadollah fight Akiyama at UFC Fight Night, siah. |

Table 19: Assessing Multi-VALUE vs. ENDIVE for translation quality across (**CollSgE**).

# F   TRANSLATION PROMPTS

---

```
Here are examples of African American Vernacular
English (AAVE):
```
1. I was bewildered, but I knew dat it was no gud asking his ass to explain.
2. Cochran pontificated windily for da camera.
3. I don't want them to follow in my footsteps, as I ain't go to no college, but I want them to go.
```
Here is the input text: {text}
Please rewrite the input text in African American
Vernacular English (AAVE).
```

---

Table 20: Few-Shot Prompt for Translating SAE to AAVE

---

```
Here are examples of Chicano English (ChcE):
```
1. When people wanna fight me I'm like "well okay, well then I'll fight you."
2. They were saying that they had a lot of problems at Garner because it was a lot of fights and stuff.
3. I ain't really thinking about getting with J. or any other guy.
```
Here is the input text: {text}
Please rewrite the input text in Chicano English
(ChcE).
```

---

Table 21: Few-Shot Prompt for Translating SAE to ChcE

---

```
Here are examples of Colloquial Singapore English
(Singlish) (CollSgE):
```
1. But after a while it become quite senseless to me.
2. And got to know this kind-hearted scholar who shelter her with \O {} umbrella when it was raining.
3. The cake John buy one always very nice to eat.
```
Here is the input text: {text}
Please rewrite the input text in Colloquial Singapore
English (Singlish) (CollSgE).
```

---

Table 22: Few-Shot Prompt for Translating SAE to CollSgE

```
Here are examples of Indian English (IndE):
```
1. It was not too much common. Getting the accommodation has become very much difficult.

2. During monsoon we get lot of rain and then gets very soggy and sultry.

3. This is the second time that such an object had been sighted here.

```
Here is the input text: {text}
Please rewrite the input text in Indian English
(IndE).
```

Table 23: Few-Shot Prompt for Translating SAE to IndE

```
Here are examples of Jamaican English (JamE):
```
1. Hill had initially been indicted with the Canute and the Michelle Saddler and their three companies.

2. The autopsy performed on Mae's torso shortly after it was found, revealed that her body was cut into pieces by a power machine saw.

3. The culture of the region has been unique in combining British and Western influences with African and Asian lifestyles.

```
Here is the input text: {text}
Please rewrite the input text in Jamaican English
(JamE).
```

Table 24: Few-Shot Prompt for Translating SAE to JamE

# G EVALUATION PROMPTS

---

```
Given a mathematics problem, determine the answer.
Simplify your answer as much as possible and
encode the final answer in <answer></answer> (e.g.,
<answer>42</answer>).
Context: {problem}
Question: {question}
Answer:
If CoT: Let's think about this step by step before
finalizing the answer.
```

---

Table 25: Prompt for SVAMP Evaluation

---

```
Given a coding problem, produce a Python function
that solves the problem.  Provide your entire code
in <answer></answer> (e.g., <answer>def solve():
pass</answer>).
Problem: {problem}
Test Cases: {test_cases}
Answer:
If CoT: Let's think step by step about the
problem-solving process before coding.
```

---

Table 26: Prompt for MBPP Evaluation

---

```
Given a yes/no question, answer yes or no.  Provide
your final answer in <answer></answer> (e.g.,
<answer>yes</answer>).
Context: {context}
Question: {question}
Answer:
If CoT: Let's think step by step before arriving at
the answer.
```

---

Table 27: Prompt for LogicBenchYN Evaluation

```
Given a multiple-choice question with 4 choices,
pick the correct choice number (1, 2, 3, or 4).
Provide your final answer in <answer></answer> (e.g.,
<answer>2</answer>).
Context: {context}
Choices:
1) {choice1}
2) {choice2}
3) {choice3}
4) {choice4}
Answer:
If CoT: Let's analyze each choice step by step before
determining the correct one.
```

Table 28: Prompt for LogicBenchMCQ Evaluation

```
Given a coding problem, produce a Python function
that solves the problem.  Provide your entire code
in <answer></answer> (e.g., <answer>def solve():
pass</answer>).
Problem: {prompt_text}
Test Cases: {test_cases}
Answer:
If CoT: Let's break the problem down step by step
before writing the code.
```

Table 29: Prompt for HumanEVAL Evaluation

```
Given a mathematics problem, determine the answer.
Simplify your answer as much as possible and
encode the final answer in <answer></answer> (e.g.,
<answer>1</answer>).
Problem: {problem}
Answer:
If CoT: Let's carefully solve the problem step by step
before arriving at the final numeric answer.
```

Table 30: Prompt for GSM8K Evaluation

```
Given premises and a conclusion, determine whether
the conclusion is True, False, or Uncertain.  Provide
your final answer in <answer></answer> (e.g.,
<answer>True</answer>).
Premises:  {premises}
Conclusion:  {conclusion}
Answer:
If CoT: Let's evaluate the premises step by step
before deciding the conclusion.
```

Table 31: Prompt for FOLIO Evaluation

```
Given a pronoun resolution problem, determine whether
Span 2 refers to Span 1.  Provide your final answer in
<answer></answer> (e.g., <answer>1</answer> for same
or <answer>0</answer> for different).
Paragraph:  {paragraph}
Span 1:  {span1}
Span 2:  {span2}
Answer:
If CoT: Let's analyze the relationship between Span 1
and Span 2 step by step before answering.
```

Table 32: Prompt for WSC Evaluation

```
Given a sentence, determine its sentiment.  Provide
your final answer in <answer></answer> (e.g.,
<answer>1</answer> for positive or <answer>0</answer>
for negative).
Sentence:  {sentence}
Answer:
If CoT: Let's analyze the sentiment of the sentence
step by step before concluding.
```

Table 33: Prompt for SST-2 Evaluation

```
Given a paragraph, a question, and an answer choice,
determine if the answer choice is correct.  Provide
your final answer in <answer></answer> (e.g.,
<answer>1</answer> for correct or <answer>0</answer>
for incorrect).
Paragraph: {paragraph}
Question: {question}
Answer Choice: {answer_choice}
Answer:
If CoT: Let's analyze the paragraph and question
step by step before confirming the correctness of the
answer choice.
```

Table 34: Prompt for MultiRC Evaluation

```
Given a premise and two choices, pick which choice
is more plausible.  Provide your final answer in
<answer></answer> (e.g., <answer>0</answer> for the
first choice or <answer>1</answer> for the second).
Premise: {premise}
Choice 1: {choice1}
Choice 2: {choice2}
Answer:
If CoT: Let's compare the plausibility of both choices
step by step before finalizing.
```

Table 35: Prompt for COPA Evaluation

```
Given a passage and a yes/no question, label it
as TRUE or FALSE. Provide your final answer in
<answer></answer> (e.g., <answer>TRUE</answer>).
Passage: {passage}
Question: {question}
Answer:
If CoT: Let's carefully consider the passage and the
question step by step before labeling the answer.
```

Table 36: Prompt for BoolQ Evaluation

## H FLUENCY SCORING PROMPT

You are an expert linguist capable of detailed
chain-of-thought reasoning.
You are given two pieces of text:
1) Original Text (SAE) { the standard American English
version.
2) Dialect Text { a translated or adapted version in
the {dialect} dialect.
Please evaluate the Dialect Text for:
1) Fluency in {dialect}:
  - Grammar, syntax, word choice, and overall
naturalness in {dialect}.
  - Consistency, flow, and readability in {dialect}.
2) Meaning Preservation:
  - Does the Dialect Text retain the same meaning or
intent as the Original Text (SAE)?
  - Are there changes or omissions that alter the
meaning?
Use the following 1{7 scoring rubric (focused on
fluency, but keep meaning in mind):
- 1:  Completely unnatural, pervasive errors, nearly
unintelligible.
- 2:  Major issues in accuracy/naturalness, very
awkward for {dialect}.
- 3:  Noticeable errors or unnatural phrasing, partial
alignment with {dialect}.
- 4:  Average fluency, some issues; mostly
understandable in {dialect}.
- 5:  Good fluency, minor errors; consistent with
{dialect}.
- 6:  Very good fluency, rare issues; flows smoothly
in {dialect}.
- 7:  Excellent fluency, fully natural, error-free,
perfectly aligned with {dialect}.
Instructions:
1.  Provide a chain-of-thought explanation comparing
meaning and evaluating fluency.
2.  End with a single line:  "Fluency Score:  X"
(where X is an integer 1{7).
Begin your detailed chain-of-thought analysis now.

Table 37: Prompt for Fluency Evaluation

# I  PREFERENCE TESTS PROMPT

```
You are an expert linguist with a strong understanding
of {dialect}.
You are given:
1) Original Text (SAE) { a standard American English
version for reference.
2) Translation A { a version in the {dialect} dialect.
3) Translation B { another version in the {dialect}
dialect.
Your task:  Decide which translation is better in the
context of the {dialect} dialect with respect to:
- Fluency (grammar, syntax, word choice, overall
naturalness in {dialect})
- Accuracy (faithfulness to the original meaning, but
expressed naturally in {dialect})
- Readability (cohesion, clarity, and flow in
{dialect})
- Cultural appropriateness (if relevant to {dialect})
Provide a detailed chain-of-thought (reasoning) as to
how you weigh these factors.
Then conclude with one final line in the exact format:
"Final preference score:  X"
(where X = 1 if you prefer Translation A, or X = 2 if
you prefer Translation B).
Make sure you reveal your full thought process, then
end with:
Final preference score:  X
```

Table 38: Prompt for Translation Comparison Evaluation

