# OpenReview forum: "Endive: A Cross-Dialect Benchmark for Fairness and Performance in Large Language Models"
_ICLR.cc/2025/Workshop/BuildingTrust — BuildingTrust_

### Official Review · Reviewer_YLh5 · 2025-02-20
**This paper introduces ENDIVE, a benchmark designed to evaluate the fairness and performance of large language models (LLMs) across underrepresented English dialects. The study is well-structured, with clear methodology and comprehensive experiments using few-shot prompting and human validation. The results demonstrate that LLMs consistently underperform on non-standard dialects compared to Standard American English (SAE), highlighting significant biases in current language technologies. The paper is original, relevant, and contributes valuable insights to the field. However, it is limited by a narrow task coverage and evaluation of only five LLMs. Overall, the paper is a strong candidate for acceptance, with potential for further exploration in future work.**

**Rating:** 7
**Confidence:** 4

**Review:**

Quality
The paper is well-structured and presents a comprehensive evaluation of large language models (LLMs) across underrepresented English dialects using the ENDIVE benchmark. The methodology is robust, combining few-shot prompting with human validation to ensure linguistic authenticity. The experiments are well-designed, and the results are clearly presented, with detailed tables and analysis. The paper also acknowledges its limitations and suggests directions for future work, which adds to its credibility.

Clarity
The paper is generally clear and well-written. The abstract provides a concise overview of the study, and the introduction effectively sets the stage for the research. The methodology section is detailed and explains the translation strategies and evaluation metrics clearly. However, some parts of the paper could benefit from more explicit explanations, particularly in the results section, where the implications of the findings could be discussed in greater depth.

Originality
The paper addresses a significant and timely issue in the field of NLP: the underperformance of LLMs on non-standard dialects. The introduction of the ENDIVE benchmark is novel and contributes to the ongoing discourse on fairness and inclusivity in language technologies. The paper builds on existing work but offers new insights into how dialectal variations can be systematically evaluated.

Significance
The findings of this paper are highly relevant to the field of NLP, particularly in the context of fairness and inclusivity. The demonstration that LLMs consistently underperform on non-standard dialects compared to Standard American English (SAE) is a valuable contribution. This work has the potential to influence future research and practical applications in the development of more equitable language technologies.

Pros
Novel Benchmark: The introduction of the ENDIVE benchmark is innovative and addresses a critical gap in NLP evaluation.

Comprehensive Evaluation: The paper presents a thorough evaluation of LLMs across multiple dialects and tasks.

Clear Results: The results are well-presented and supported by detailed tables and analysis.

Future Work: The paper acknowledges its limitations and suggests valuable directions for future research.

Cons
Limited Task Coverage: The benchmark focuses on 12 tasks, which may not cover all aspects of dialectal variation.

Model Coverage: The study evaluates only five LLMs, which may not be representative of the rapidly evolving field.

Depth of Discussion: The implications of the findings could be discussed in greater depth, particularly in relation to existing literature and potential real-world applications.

---

### Official Review · Reviewer_oQ1v · 2025-02-25
**Timely work, sound methodology**

**Rating:** 7
**Confidence:** 4

**Review:**

Given the improvement in LLM performance across all languages, it is a good time to also focus on dialects -- performance on which might be affected by the model's knowledge in the corresponding non-dialect language. This work investigates this, focusing on five English dialects to profile the performance of language models on them. The methods are clearly explained, the research is timely.

Strengths:
- Inclusive and generally exhaustive. 5 English dialects. 4 task types. 12 datasets. Zero shot and CoT inference. Standard stuff. Could always be more exhaustive, but this is good enough.
- Proper automatic translation and filtering methods, which seem correct to me, with various scores to quality-check them: diversity scores, fluency scores, translation scores.
- Representative selection of models for today's state of art, even if it moves very quickly.

Possible improvements:
- More variety in dialects: English is already a high resource language, and most of the dialects are perhaps too similar to generalize the results to LLMs' performance in dialect
- Most models here have degraded performance on the dialects, the beginnings of an exploratory analysis would be great: Why does this happen? Are there certain words or phrases in certain dialects that cause this? Do these words/phrases share commonalities across dialects? Says in pg 7 that the models face challenges "in coreference resolution and textual comprehension" for the dialects, but no further explanation given, so perhaps the work would benefit a better qualitative analysis.

The work doesn't explicitly attempt to be sociological, but it might be a good idea to provide markers that lead to sociological research and to scratch the surface of why dialects suffer. But again, very timely, meaningful work. A little top-heavy on the experimental setup and filtering, could use more qualitative analyses to balance it out.

---

### Official Review · Reviewer_mgNd · 2025-03-02
**The paper makes a contribution by providing a detailed and large-scale benchmark for non-standard English dialects, nicely backed up with quality tests for the few-shot translation approach.**

**Rating:** 7
**Confidence:** 2

**Review:**

## Summary
The paper proposes a benchmark for evaluating model performance on non-standard English dialects across tasks in language understanding, algorithmic reasoning, mathematics, and logic. The authors introduce a method for translating questions from Standard American English to dialects using few-shot prompting with three verified examples informed by eWAVE. Five different LLMs are tested on this benchmark, often showing lower performance, demonstrating a drop in capability when models are prompted in non-standard English dialects.

## Strengths
- This benchmark includes five different non-standard dialects, unlike related work that often focuses on a single dialect. The translation methodology appears easily scalable to more dialects.
- The evaluated LLMs are recent and widely used models.
- The authors conducted multiple experiments to assess translation quality, including BLEU score filtering, lexical diversity, BARTScore, fluency evaluation, and human ratings.
- The benchmark covers a wide variety of tasks in both zero-shot and CoT formats.
- The authors clearly outline the limitations of their proposed benchmark.

## Weaknesses/Unclear Points
- I could not find a full example of the translated data. Based on the translation prompt, it seems the formality of examples may decrease after translation, making performance comparisons to the original questions unfair, as the original questions are often in a very formal style. Although human feedback rates formality highly, I would still appreciate seeing full examples after translation.
- Is there a reason why fluency is not tested for Multi-VALUE?
- On line 176, the Appendix number is missing.

Overall, the paper makes a valuable contribution by providing a detailed and large-scale benchmark for non-standard English dialects.

---

### Decision · Program_Chairs · 2025-03-04

Accept